# Targeted Suicide Gene Therapy with Retroviral Replicating Vectors for Experimental Canine Cancers

**DOI:** 10.3390/ijms25052657

**Published:** 2024-02-24

**Authors:** Emiko Sonoda-Fukuda, Yuya Takeuchi, Nao Ogawa, Shunsuke Noguchi, Toru Takarada, Noriyuki Kasahara, Shuji Kubo

**Affiliations:** 1Laboratory of Molecular and Genetic Therapeutics, Institute of Advanced Medical Science, Hyogo Medical University, Nishinomiya 663-8501, Japan; 2Departments of Biomedical Chemistry, School of Science and Technology, Kwansei Gakuin University, Sanda 669-1330, Japan; 3Laboratory of Veterinary Radiology, Graduate School of Veterinary Science, Osaka Metropolitan University, Izumisano 598-8531, Japan; 4Laboratory of Functional Molecular Chemistry, Kobe Pharmaceutical University, Kobe 658-8558, Japan; 5Departments of Neurological Surgery and Radiation Oncology, University of California, San Francisco, CA 94143, USA

**Keywords:** canine cancer, retroviral replicating vectors, suicide gene therapy

## Abstract

Cancer in dogs has increased in recent years and is a leading cause of death. We have developed a retroviral replicating vector (RRV) that specifically targets cancer cells for infection and replication. RRV carrying a suicide gene induced synchronized killing of cancer cells when administered with a prodrug after infection. In this study, we evaluated two distinct RRVs derived from amphotropic murine leukemia virus (AMLV) and gibbon ape leukemia virus (GALV) in canine tumor models both in vitro and in vivo. Despite low infection rates in normal canine cells, both RRVs efficiently infected and replicated within all the canine tumor cells tested. The efficient intratumoral spread of the RRVs after their intratumoral injection was also demonstrated in nude mouse models of subcutaneous canine tumor xenografts. When both RRVs encoded a yeast cytosine deaminase suicide gene, which converts the prodrug 5-fluorocytosine (5-FC) to the active drug 5-fluorouracil, they caused tumor-cell-specific 5-FC-induced killing of the canine tumor cells in vitro. Furthermore, in the AZACF- and AZACH-cell subcutaneous tumor xenograft models, both RRVs exerted significant antitumor effects. These results suggest that RRV-mediated suicide gene therapy is a novel therapeutic approach to canine cancers.

## 1. Introduction

In recent years, the life expectancy of dogs has increased and cancer is becoming more common. Canine cancer is now a major cause of death in dogs, accounting for approximately one-third to half of all deaths in older dogs [1,2,3]. The current first-line treatment for canine cancer is surgery, followed by chemotherapy, radiation therapy, and immunotherapy, including immune checkpoint inhibitors [1,3]. However, there are still several concerns in treating canine cancers, including the lack of experience in treating dogs compared with humans, the diversity of dog breeds, and the limited medical facilities available for dogs. Therefore, the development of effective treatment methods for canine cancers is required.

Oncolytic virotherapy has been attracting attention as a novel therapeutic option for human cancers [2,4,5]. Human clinical trials have been conducted of oncolytic viruses created from cytotoxic viruses, including adenovirus [6], herpes simplex virus [7,8,9], and reovirus [10]. These are genetically modified to exclusively target cancer cells for infection and replication or do so naturally. Canine cancer has also been treated with this novel therapy, and successful virotherapies for canine tumors have been reported based on canine adenovirus [11], reovirus [12], Sendai virus [13], canine distemper virus [14], and Newcastle disease virus [15]. However, because these viruses are highly immunogenic and are expected to induce tumor immunity, problems arise from the early elimination of infected cells by the host immune system, hindering sufficient intratumor spread, and leading to limited or transient efficacy and possibly tumor regrowth [4,16].

Taking a different approach from previous oncolytic viruses, we created retroviral replicating vectors (RRVs) that were based on the amphotropic murine leukemia virus (AMLV) and the gibbon ape leukemia virus (GALV) [17,18]. Both RRVs naturally and specifically infect human cancer cells, because most cancer cells lack innate immunity against retroviruses and the environment within the tumor is immunoprivileged [17,19]. Therefore, the RRVs will undergo multiple rounds of replication and spread throughout the tumor, increasing the proportion of infected cells and infected areas overall, including the tumors that were not injected. Second, the viral genomes also integrate steadily into the genomes of the infected tumor cells during the replication process, generating a reservoir of RRVs. This reservoir acts as a source of RRV production and RRVs spread to newly arising cancer cells. Third, RRVs are not cytotoxic in themselves, unlike other oncolytic viruses. To exert its antitumor effect, the RRV is armed with a suicide gene that induces synchronized tumor-cell death when administered with the prodrug, allowing the continued destruction of tumor cells after each of multiple cycles of intermittent prodrug administration. Using RRVs armed with the yeast cytosine deaminase (CD) suicide gene, we demonstrated their highly effective killing of various cancer cells in vitro and in vivo [17,18,20,21,22,23]. Human clinical trials of RRV-mediated suicide gene therapy have begun, demonstrating its therapeutic benefit in treating recurrent high-grade glioma [24,25].

In this study, we assessed the cancer-selective, highly efficient gene delivery and therapeutic efficacy of two different RRVs, AMLV and GALV, in canine tumor cells, both in vitro and in vivo.

## 2. Results

### 2.1. RRVs Replicate and Spread Efficiently in Canine Cancer Cell Lines

First, we analyzed the replication kinetics of AMLV–GFP and GALV–GFP (Figure 1) in 10 canine cancer cell lines (AZACF, AZACH, AZACL1, AZACL2, AZACB, AZACU, oSCC-1, oSCC-4, TSCCLN#1, and TSCCLN#4) and two normal canine cells (fibroblasts and liver epithelial cells).

Neither of these RRVs infected or spread in either of the normal cells (Figure 2). However, both RRVs effectively infected and spread in all the cancer cell lines. In particular, both RRVs showed more than 90% infection efficiency in AZACF fibrosarcoma cells within 10 days of infection. Remarkably, GALV spread more quickly than AMLV in AZACH hepatoma cells. These results indicate that these RRVs replicated and spread efficiently in a cancer-cell-specific manner in the canine cells used in this study.

### 2.2. RRVs Replicate and Spread Efficiently in Subcutaneous Canine Xenograft Tumors in Nude Mice

To evaluate the spread of the RRVs in vivo, nude mice with subcutaneous xenografts of three tumor cell lines (AZACF, AZACH, and AZACL2) were injected intratumorally with firefly luciferase (luc)-expressing vector AMLV–luc or GALV–luc, or the replication-deficient lentiviral vector (LV–luc) as the control, and the luminescent signal was analyzed with bioluminescence imaging on the indicated days (Figure 3a,b). In all the control group tumors, the signal was first seen on day 3, and grew in proportion to the tumor size. In contrast, in both the RRV-treated groups, the signals increased strongly and had spread throughout the entire tumor by day 28, with no signal in the normal extratumoral tissue (Figure 3a).

Quantification of the luminescence intensity in the entire tumors revealed that both RRV-treated groups had lower levels of signal at first than the control group, but that their signals increased quickly day by day, until they plateaued and exceeded the levels of the control group (Figure 3b). Remarkably, GALV spread more quickly in AZACH-cell tumors than AMLV, which is consistent with the findings of the in vitro study on the replication kinetics of RRV in AZACH cells (Figure 2). These results suggest that the RRVs effectively replicate and spread through canine tumors in vivo.

### 2.3. Suicide-Gene-Mediated Tumor-Cell-Specific Killing Effect of RRVs on Canine Cancer Cells

We then assessed the cell-killing effect of RRV–CD, which expressed the CD suicide gene (Figure 1), combined with the prodrug 5-fluorocytosine (5-FC) on two normal canine cells (fibroblasts and liver epithelial cells) and five canine cancer cells (AZACF, AZACH, AZACL2, oSCC-4, and TSCCLN#4) (Figure 4). In the normal cells, neither RRV–CD (GALV or AMLV) showed any cell-killing activity with 10 mM 5-FC. In contrast, the viability of the cancer cells was <10% with 10 mM 5-FC, and the cell-killing effects of both RRV–CDs on cancer cells were largely dependent on the 5-FC concentration. These results show that RRV–CD exerts the cell-killing effect of the suicide gene without harming normal cells. Furthermore, GALV showed superior efficacy to AMLV in eliminating AZACH cells, consistent with the different abilities of the RRVs to replicate and spread in these cells (Figure 2 and Figure 3).

### 2.4. RRV–CD/5-FC Suicide Gene Therapy Exerts Potent Antitumor Effects on Subcutaneous Canine Xenograft Mouse Models

To examine the potential antitumor effects of RRV-mediated CD/5-FC suicide gene therapy, nude mice bearing subcutaneous tumors (AZACF or AZACH) were intratumorally administered AMLV–CD, GALV–CD, or phosphate-buffered saline (PBS; as the control). 5-FC was subsequently administered intraperitoneally three times a week. As shown in Figure 5 (left column), both RRVs showed potent antitumor effects in the AZACF model (*p* = 0.0002 for AMLV and *p* < 0.0001 for GALV vs. PBS group on day 31). However, in the AZACH model (right column), GALV had a greater anticancer effect than AMLV (*p* = 0.001 on day 49), although both RRVs exerted significant antitumor effects compared with that in the PBS-treated group (*p* < 0.0001 for AMLV and <0.0001 for GALV on day 49). These results are consistent with the replication efficiencies (Figure 2 and Figure 3) and killing effects (Figure 4) of the RRVs in AZACH cells, which demonstrated the dominance of GALV over AMLV. These results indicate that RRV-mediated suicide gene therapy has a significant antitumor effect on canine tumors. Moreover, the efficiency of RRV replication and spread determines this efficacy.

## 3. Discussion

In this study, we showed that RRVs efficiently infect and spread in a variety of canine cancer cells, both in vitro and in vivo. We also showed highly effective cancer-cell-selective cytotoxicity against canine cancer cells in vitro using RRV expressing the CD suicide gene and an anticancer effect in vivo after the administration of the prodrug 5-FC. This is the first preclinical study to suggest that RRV-mediated suicide gene therapy is a potentially effective cancer treatment for dogs.

In the canine cancer cells examined in this study, both RRVs efficiently replicated and spread, regardless of the tumor type (Figure 2). Surprisingly, both RRVs demonstrated more effective spread and a stronger cell-killing effect in canine cancer cells than in human cancer cells in our earlier research [18,26,27]. Therefore, RRV-mediated suicide gene therapy may be equally effective as—or even more effective than—the methods currently used to treat cancer in humans, although further research is required. However, because bloodlines are important in dog breeding, there may be more genetic diversity across dog breeds than among humans [28], and this diversity could affect their response to treatment. Dogs with diverse genetic backgrounds are expected to benefit greatly from cancer virotherapy, which is generally effective regardless of the genetic background. If such a breakthrough therapy is developed, it could greatly improve the prognosis of dogs with cancer, independently of their genetic background. While such a treatment would be valuable in canine cancer care, it may also offer human applications. Therefore, our study represents a noteworthy advance, contributing to both veterinary and human medical sciences.

As noted in the Introduction, RRV replicates in tumor cells and is incorporated into the tumor-cell genome, creating a reservoir of RRV and allowing RRV to remain in the tumor for a long time. RRV armed with the CD suicide gene induces cancer cell killing with the synchronous administration of 5-FC, and allows the ongoing destruction of tumor cells, disrupting immunological resistance and reactivating the immune system against the tumor. According to preclinical models of this phenomenon, multiple 5-FC cycles are required to elicit a sustained response in immunocompetent animals [29]. In other words, these unique characteristics of RRVs make it possible to target cancers with long-term control by maximizing the dose of prodrug administered, its duration, and the interval of administration [17,30,31]. Therefore, RRV-mediated suicide gene therapy is a controlled regimen designed to prolong patient life and lessen suffering by inhibiting the growth and spread of a cancer, rather than eliminating it hastily.

Current viral therapies, including RRV, are generally administered intratumorally. The therapies are readily applicable and likely to show their true worth in actual clinical practice when treating canine tumors that are discovered by palpation or examination and are located close to the exterior surface. The prodrug 5-FC can be administered orally, facilitating treatment on an outpatient basis after a single injection of RRV into the tumor. This is a promising feature that would shorten hospital stays, thus increasing bed occupancy in the limited advanced medical facilities available for dogs, and improve the quality of life of canine patients.

The two different RRVs, AMLV and GALV, utilize different isoforms of the type III phosphate transporter as receptors, PiT-2 (SLC20A2) and PiT-1 (SLC20A1), respectively [32,33,34]. These phosphate transporters are overexpressed more often in tumor cells than in normal cells, and the efficiency of RRV spread and their efficacy in human tumors are dependent upon the expression levels of these transporters [18,26]. Therefore, personalized virotherapy is anticipated based on the expression of cellular receptors in biopsy specimens from human tumors. In this context, elucidating the expression pattern of RRV receptors in canine tumor cells is an important approach for future research. Notably, the present study showed that the dominance of GALV over AMLV in terms of replication efficiency, in vitro killing efficacy, and in vivo antitumor effect was demonstrated in AZACH cells. However, the expression of RRV receptors in canine tumor cells has not yet been determined due to the low homology of the PiT-1/PiT-2 genes between dogs and humans, and the greater genetic differences among canine breeds than among humans. We will continue to work to resolve this issue.

In this study, we investigated the efficacy of suicide gene therapy for experimental canine tumors using RRVs derived from AMLV and GALV. This is the first preclinical study to demonstrate that RRV-mediated suicide gene therapy holds promise for the treatment of canine cancers and has potential applications in the veterinary field.

## 4. Materials and Methods

### 4.1. Cell Culture

All cells used are listed in Table 1. Normal canine fibroblasts were isolated from the subcutaneous tissue of a miniature dachshund’s head, and were provided by Dr. Noguchi (Osaka Metropolitan University, Izumisano, Japan). Normal canine liver epithelial cells were purchased from Cell Biologics, Inc. (Chicago, IL, USA). Canine tumor cell lines (AZACF fibrosarcoma cells, AZACH hepatoma cells, AZACL1 and AZACL2 lung carcinoma cells, AZACB breast tumor cells, and AZACU urothelial carcinoma cells) were purchased from Cosmo Bio Co., Ltd. (Tokyo, Japan). Maxillary squamous carcinoma cells (oSCC-1 and oSCC-4) and tonsillar squamous carcinoma cells (TSCCLN#1 and TSCCLN#4) were provided by Dr. Noguchi [35].

The normal canine fibroblasts and AZACF, AZACL1, AZACL2, AZACB, AZACU, oSCC-1, oSCC-4, TSCCLN#1, and TSCCLN#4 cells were grown in Roswell Park Memorial Institute (RPMI) 1640 medium (Nacalai Tesque, Kyoto, Japan) supplemented with 10% fetal bovine serum (FBS; Atlas Biologicals, Fort Collins, CO, USA). AZACH cells were grown in RPMI 1640 medium supplemented with 3% FBS. Normal liver epithelial cells were grown in epithelial cell medium purchased from Cell Biologics, Inc. on gelatin-coated dishes and plates (Corning Inc., Corning, NY, USA). 293T cells were cultured in Dulbecco’s modified Eagle’s medium (Nacalai Tesque) supplemented with 10% FBS. All cells were cultured at 37 °C under 5% CO_2_.

### 4.2. Viral Vector Plasmids and Virus Production

The AMLV and GALV vector plasmids were produced as described previously (Figure 1) [18,22,28,29]. Briefly, pAMLV–GFP and pGALV–GFP contained full-length replication-competent amphotropic AMLV and GALV proviruses, respectively, with internal ribosome entry site and green fluorescent protein (GFP) coding sequences. pAMLV–CD and pGALV–CD were created from the plasmids described above, respectively, by replacing the GFP gene with a CD suicide gene, and pAMLV–luc and pGALV–luc were constructed with a firefly luciferase gene (luc) (Figure 1).

To produce RRVs, 293T cells were transfected with each individual plasmid using Lipofectamine 2000 (Life Technologies Japan, Tokyo, Japan). At 48 h after transfection, the virus-containing supernatant was collected, filtered, and stored at −80 °C [18,22,28,29]. The titers of the RRV–GFPs were determined based on GFP expression, with a FACSCalibur flow cytometer (Becton Dickinson Japan, Tokyo, Japan), and expressed as transduction units per mL. All vectors were prepared in parallel and confirmed to be of similar titers with quantitative real-time PCR.

To produce the self-inactivating replication-deficient lentiviral vector expressing firefly luciferase (LV–luc), 293T cells were cotransfected with pLV–luc, pMDLg/p (encoding gag–pol), pRSV-Rev (encoding rev), and pMD.G (encoding VSV-G env) [18,21]. At 48 h after transfection, the virus-containing supernatant was collected, filtered, concentrated with ultracentrifugation, and stored at −80 °C.

### 4.3. Analysis of RRV Replication Kinetics In Vitro

Cells at 20–30% confluence were infected with AMLV–GFP or GALV–GFP at a multiplicity of infection (MOI) of 0.01. At a series of time points, the cells were trypsinized, and the detached cells were resuspended in PBS and analyzed with a FACSCalibur flow cytometer. Ten thousand cells per sample were analyzed, and the remaining cells were cultured and passaged.

### 4.4. RRV Replication Kinetics in a Subcutaneous Tumor Model

Female BALB/c-nu/nu (nude) mice (CLEA Japan, Tokyo, Japan) were maintained under specific-pathogen-free conditions, and all experiments were performed under protocols approved by the Animal Research Committee of Hyogo Medical University. Subcutaneous tumor models were established by subcutaneously inoculating 5-week-old female mice with AZACF, AZACH, or AZACL2 xenografts containing 1 × 10^6^ tumor cells on the right dorsal surface. The vectors LV–luc (1 µL), AMLV–luc (10 µL), and GALV–luc (10 µL) were diluted to 100 µL with Dulbecco’s PBS and injected into the tumors when the long diameter of the tumor was >5 mm (day 0). In vivo imaging was performed with IVIS Lumina II (Caliper Life Sciences Inc., Hopkinton, MA, USA) after the intraperitoneal injection of D-luciferin (150 µg/g bodyweight; Biosynth, Compton, UK) while the mice were anesthetized with isoflurane inhalation (Mylan Inc., Canonsburg, PA, USA). To quantify the luminescence intensity in the tumors, the signal in the tumor area was analyzed in all images taken with the ROI tool in the Living Image software ver. 4.2 (Caliper Life Sciences Inc.).

### 4.5. In Vitro Cytotoxicity Assay

Cells at 20–30% confluence were infected with AMLV–CD or GALV–CD at an MOI of 0.01. On day 15, 5-FC (Tokyo Chemical Industry, Tokyo, Japan) was added to the culture medium at final concentrations of 0.1, 1, or 10 mM. On day 21, the percentage of viable cells was determined with alamarBlue (Thermo Fisher Scientific Inc., Waltham, MA, USA), in accordance with the manufacturer’s instructions. Briefly, 10% alamarBlue reagent was added aseptically to the culture medium, which was then incubated for 3 h. Fluorescence was measured with an Infinite M200 PRO microplate reader (Tecan Group Ltd., Männedorf, Zurich, Switzerland) at an excitation wavelength of 544 nm and a fluorescence wavelength of 590 nm. The percentage of viable cells was determined by calculating the fluorescence of viable cells measured against wells without 5-FC.

### 4.6. Subcutaneous Xenograft Models of Canine Tumors

Female BALB/c-nu/nu (nude) mice (The Jackson Laboratory Japan, Inc., Kanagawa, Japan) were maintained under specific-pathogen-free conditions and all experiments were performed according to protocols approved by the Animal Research Committee of Hyogo Medical University. Four-week-old female mice were inoculated subcutaneously with AZACF and AZACH xenografts of 1 × 10^6^ tumor cells on the right dorsal surface to establish subcutaneous tumor models. When the tumor long diameter was >5 mm (day 0), each model was divided into three groups, and 10 µL of AMLV–CD or GALV–CD was diluted with PBS to 100 µL and injected into the tumors. PBS (100 µL) was injected into the control group. When the tumor volume exceeded 100 mm^3^ (days 3 and 14), 5-FC was administered intraperitoneally three times a week at a dose of 500 µg/g bodyweight/day. The tumor volume was measured three times a week and calculated as V = longest diameter × (shortest diameter)^2^ × 0.5. Each group consisted of at least eight mice for AZACF xenografts and 12 mice for AZACH xenografts.

### 4.7. Statistical Analysis

Data are presented as means ± standard errors (SE), and were analyzed with Student’s *t* test. A *p*-value of <0.05 indicated a significant difference.

## Figures and Tables

**Figure 1 ijms-25-02657-f001:**
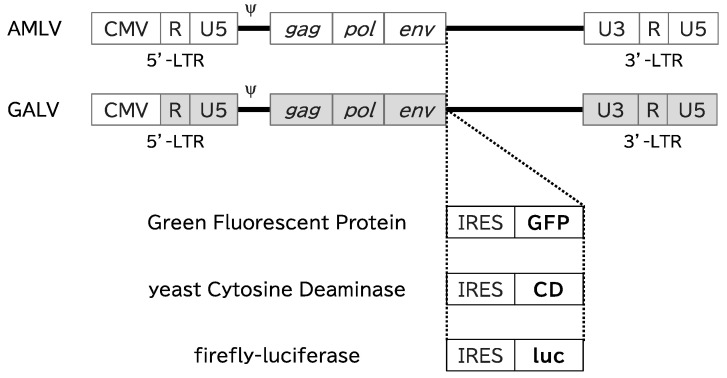
Schematic structure of RRVs. Each RRV contains the full-length replication-competent AMLV or GALV provirus. An internal ribosomal entry site (IRES) and gene of interest (GFP, CD, or luciferase) were inserted between the env gene and the 3′-untranslated region. CMV: cytomegalovirus-derived promoter; LTR: long terminal repeat sequence, ψ: packaging signal; *gag/pol/env*: retroviral structural genes; GFP: green fluorescent protein; CD: yeast cytosine deaminase, luc: firefly luciferase.

**Figure 2 ijms-25-02657-f002:**
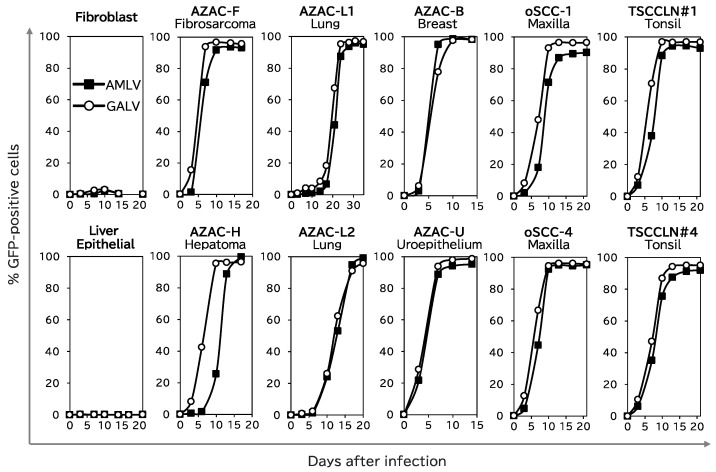
Replication kinetics of RRVs in canine cells. Normal canine cells (fibroblasts and primary liver epithelial cells) and canine tumor cell lines (AZACF, AZACH, AZACL1, AZACL2, AZACB, AZACU, oSCC-1, oSCC-4, TSCCLN#1, and TSCCLN#4) were infected with GFP-expressing RRV (AMLV or GALV) at a multiplicity of infection of 0.01. Cell growth was maintained and the percentage of cells expressing GFP was analyzed with a flow cytometer at every passage. Representative data from three independent experiments are shown.

**Figure 3 ijms-25-02657-f003:**
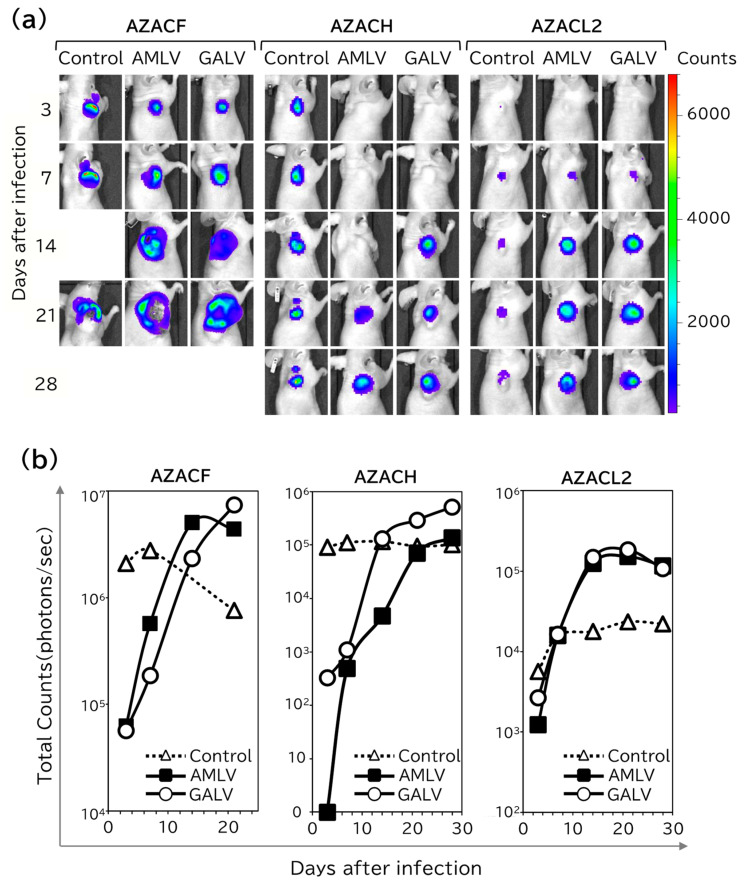
In vivo replicative spread of RRV in canine tumor xenograft models. Nude mice were inoculated under the back skin with AZACF, AZACH, and AZACL2 cells to create subcutaneous tumor models. Luc-expressing RRVs (AMLV or GALV) were administered intratumorally and observed with the IVIS imaging system from day 3 to day 28. A replication-defective lentiviral vector was used to construct the controls. (**a**) In vivo fluorescence imaging. Representative images are shown for each group. The scale indicates the numerical value of the relative luminescence units on a CCD camera. (**b**) Luminescent signal intensities in the tumors were quantified with the ROI tool in the Living Image software.

**Figure 4 ijms-25-02657-f004:**
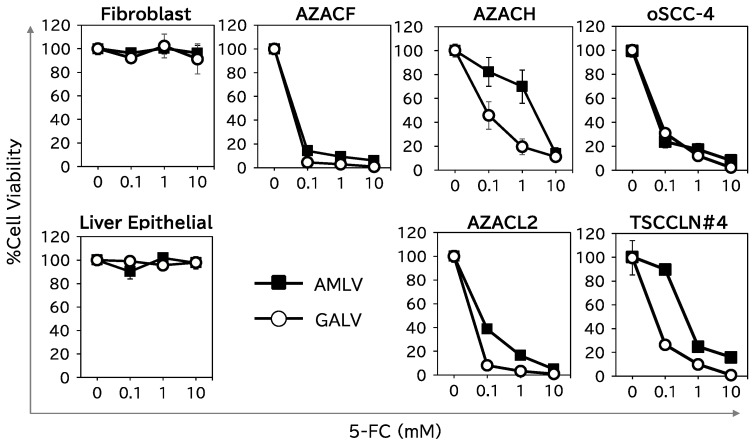
Suicide-gene-mediated cell-killing effects in canine cells. Fibroblasts, liver epithelial cells, and AZACF, AZACH, AZACL2, oSCC-4, and TSCCLN#4 cells were infected with CD-expressing RRVs (AMLV or GALV) at an MOI of 0.01. On day 15, the cells were incubated with various concentrations of 5-FC for 6 days, and then cell viability was evaluated with an alamarBlue assay. Data shown are means ± SEs of triplicate experiments.

**Figure 5 ijms-25-02657-f005:**
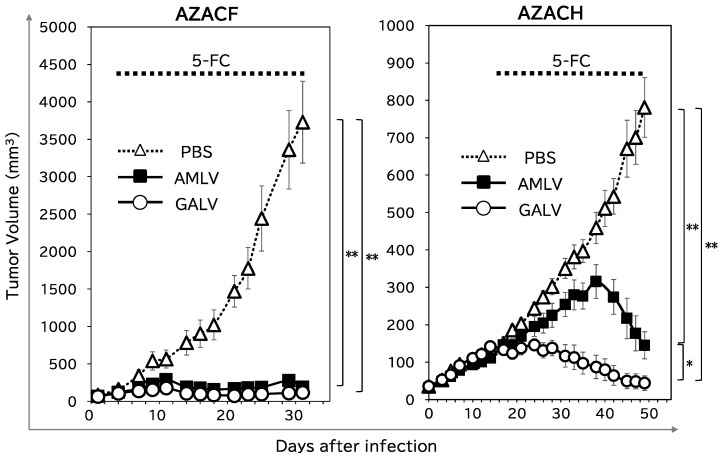
Suicide-gene-mediated antitumor effects in a canine tumor xenograft model. The backs of nude mice were subcutaneously inoculated with AZACF or AZACH cells, and CD-expressing RRVs (or PBS for the control group) were administered intratumorally when the tumor diameter reached 5 mm. 5-FC (500 µg/g bodyweight/day) was then administered intraperitoneally three times a week, commencing when the tumor volume exceeded 100 mm^3^ (AZACF: from day 3; AZACH: from day 14). Tumor volume was measured three times a week. Data shown are the means ± SEs of triplicate experiments. * *p* < 0.05, ** *p* < 0.01.

**Table 1 ijms-25-02657-t001:** List of cells used.

Cell Names	Origin	Breed	Age (Year)	Source
Fibroblast	Head skin	Miniature Dachshund	11	Dr. Noguchi (Osaka Metropolitan University)
Liver Epithelial	Liver	Beagle	U/N	Cell Biologics, Inc.
AZACF	Fibrosarcoma	U/N	U/N	Cosmo Bio Co., Ltd.
AZACH	Hepatoma	Maltese	U/N	Cosmo Bio Co., Ltd.
AZACL1	Lung Carcinoma	Shetland Sheepdog	U/N	Cosmo Bio Co., Ltd.
AZACL2	Lung Carcinoma	U/N	U/N	Cosmo Bio Co., Ltd.
AZACB	Breast Tumor	U/N	U/N	Cosmo Bio Co., Ltd.
AZACU	Urothelial Carcinoma	U/N	U/N	Cosmo Bio Co., Ltd.
oSCC-1	Maxillary squamous carcinoma	Toy Poodle	10	Dr. Noguchi (Osaka Metropolitan University) [35]
oSCC-4	Maxillary squamous carcinoma	Toy Poodle	10	Dr. Noguchi (Osaka Metropolitan University) [35]
TSCCLN#1	Tonsillar squamous carcinoma	Miniature Schnauzer	13	Dr. Noguchi (Osaka Metropolitan University) [35]
TSCCLN#4	Tonsillar squamous carcinoma	Miniature Schnauzer	13	Dr. Noguchi (Osaka Metropolitan University) [35]

U/N: Unknown.

## Data Availability

The data presented in this study are available upon request from the corresponding author.

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
