# Peer review of "Targeted Suicide Gene Therapy with Retroviral Replicating Vectors for Experimental Canine Cancers"

_ijms, 2024, doi:10.3390/ijms25052657_

Round 1
Reviewer 1 Report
Comments and Suggestions for Authors
In the presented manuscript, the authors have developed a retroviral replicating vector (RRV) that specifically targets dog cancer cells for infection and replication. RRV carrying a suicide gene induced synchronized killing of cancer cells when administered with a prodrug after infection. In the presented study, furthermore they evaluated two distinct RRVs derived from amphotropic murine leukemia virus (AMLV) and gibbon ape leukemia virus (GALV), in canine tumor models both in vitro and in vivo. Despite low infection rates in normal canine cells, both RRVs efficiently infected and replicated within all the canine tumor cells tested. The efficient intratumoral spread of the RRVs after their intratumoral injection was also demonstrated in nude mouse models of subcutaneous canine tumor xenografts. According to the authors, this is the first preclinical study to demonstrate that RRV-mediated suicide gene therapy holds promise for the treatment of canine cancers. I think that in the near future, like it is stated by the authors, it can be even a valuable therapy for humans. In general, the paper is well structured and the results are properly presented, while I would suggest to include some additional sentences in the conclusions. I would suggest to the Editor of the Journal to accept the manucript in the presented form.
Author Response
We would like to thank the Reviewer for the assessment that “I would suggest to the Editor of the Journal to accept the manucript in the presented form.”.
“…, while I would suggest to include some additional sentences in the conclusions.”
This appears to be a misunderstanding, but was probably caused by the fact that the Discussion and Conclusion statements were independent and far apart. To resolve this, the Conclusion was moved to the end of the Discussion section.
"Please see the attachment."

Reviewer 2 Report
Comments and Suggestions for Authors
The manuscript by Emiko Sonoda-Fukuda et al., entitled “Targeted suicide gene therapy with retroviral replicating vectors for experimental canine cancers” developed a novel gene therapy strategy for canine cancers. The approach is interesting. However, few concerns listed below need to be addressed. The manuscript can be accepted after a thorough revision.
1. The expression of cytosine deaminase should be quantified in tumors with western blot and qPCR.
2. In therapeutic evaluation, control virus (Virus without cytidine deaminase) group is missing. It is critical to understand whether any additional effect with the virus on the tumors through activating the immune system.
3. Since the viruses are replicative in nature, there is a high possibility to develop blood abnormalities. Hence, it is important to evaluate their effect of hematopoietic system.
Author Response
We would like to sincerely thank the Reviewer for recognizing that “The approach is interesting.”. Answers to the few concerns raised by the Reviewer are listed below.
“1. The expression of cytosine deaminase should be quantified in tumors with western blot and qPCR.”
Our previous study on CD expression in tumors after intratumor administration of RRV-CD was substantiated by semiquantitative PCR[1]. In addition, our studies to date indicate that 5-FC-dependent induction of tumor cell death is not seen with RRV-GFP, but only with RRV-CD[2,3]. This makes us confident that CD is expressed functionally, albeit indirectly. For this reason, we have not directly examined CD expression in our recent papers.
“2. In therapeutic evaluation, control virus (Virus without cytidine deaminase) group is missing. It is critical to understand whether any additional effect with the virus on the tumors through activating the immune system.”
As we have already reported, tumor growth in the RRV-GFP group with or without 5-FC is similar to that in the PBS control group and the 5-FC alone group (no RRV) [2-4]. In addition, administration of RRV-CD alone (no 5-FC) caused tumor growth similar to that in the PBS control group. In other words, it has already been proven that RRV itself has no anti-tumor effect [2,4-6]. Therefore, we did not prepare a control virus group for this study.
“3. Since the viruses are replicative in nature, there is a high possibility to develop blood abnormalities. Hence, it is important to evaluate their effect of hematopoietic system.“
The Reviewer raises an important point. As mentioned in the introduction, RRVs naturally and specifically infect human cancer cells. In addition, when administered intratumorally in xenograft rodent models of human tumor, spread of RRV to systemic organs is rare [5,7]. Even if there are normal cells infected by the RRV-CD to be administered, 5-FC administration is expected to eliminate them and reduce cancer development. Furthermore, more than 350 individuals who received RRV/5-FC in human clinical trials did not develop cancer from the vector, and molecular research revealed no potentially harmful integration sites[8-10]. However, the safety of RRV for dogs will be evaluated in more detail in future studies.
References
- Tai, C.K.; Wang, W.J.; Chen, T.C.; Kasahara, N. Single-shot, multicycle suicide gene therapy by replication-competent retrovirus vectors achieves long-term survival benefit in experimental glioma. Mol Ther 2005, 12, 842-851, doi:10.1016/j.ymthe.2005.03.017.
- Inoko, K.; Hiraoka, K.; Inagaki, A.; Takahashi, M.; Kushibiki, T.; Hontani, K.; Takano, H.; Sato, S.; Takeuchi, S.; Nakamura, T.; et al. Therapeutic activity of retroviral replicating vector-mediated prodrug activator gene therapy for pancreatic cancer. Cancer Gene Ther 2018, 25, 184-195, doi:10.1038/s41417-018-0020-7.
- Kawasaki, Y.; Tamamoto, A.; Takagi-Kimura, M.; Maeyama, Y.; Yamaoka, N.; Terada, N.; Okamura, H.; Kasahara, N.; Kubo, S. Replication-competent retrovirus vector-mediated prodrug activator gene therapy in experimental models of human malignant mesothelioma. Cancer Gene Ther 2011, 18, 571-578, doi:10.1038/cgt.2011.25.
- Kushiya, H.; Hiraoka, K.; Suzuki, T.; Inoko, K.; Inagaki, A.; Niwa, H.; Sasaki, K.; Nakamura, T.; Tsuchikawa, T.; Shichinohe, T.; et al. Retroviral Replicating Vector Toca 511 (Vocimagene Amiretrorepvec) for Prodrug Activator Gene Therapy of Lung Cancer. Cancers (Basel) 2022, 14, doi:10.3390/cancers14235820.
- Hiraoka, K.; Inagaki, A.; Kato, Y.; Huang, T.T.; Mitchell, L.A.; Kamijima, S.; Takahashi, M.; Matsumoto, H.; Hacke, K.; Kruse, C.A.; et al. Retroviral replicating vector-mediated gene therapy achieves long-term control of tumor recurrence and leads to durable anticancer immunity. Neuro Oncol 2017, 19, 918-929, doi:10.1093/neuonc/nox038.
- Hiraoka, K.; Kimura, T.; Logg, C.R.; Tai, C.K.; Haga, K.; Lawson, G.W.; Kasahara, N. Therapeutic efficacy of replication-competent retrovirus vector-mediated suicide gene therapy in a multifocal colorectal cancer metastasis model. Cancer Res 2007, 67, 5345-5353, doi:10.1158/0008-5472.CAN-06-4673.
- Ostertag, D.; Amundson, K.K.; Lopez Espinoza, F.; Martin, B.; Buckley, T.; Galvao da Silva, A.P.; Lin, A.H.; Valenta, D.T.; Perez, O.D.; Ibanez, C.E.; et al. Brain tumor eradication and prolonged survival from intratumoral conversion of 5-fluorocytosine to 5-fluorouracil using a nonlytic retroviral replicating vector. Neuro Oncol 2012, 14, 145-159, doi:10.1093/neuonc/nor199.
- Cloughesy, T.F.; Landolfi, J.; Hogan, D.J.; Bloomfield, S.; Carter, B.; Chen, C.C.; Elder, J.B.; Kalkanis, S.N.; Kesari, S.; Lai, A.; et al. Phase 1 trial of vocimagene amiretrorepvec and 5-fluorocytosine for recurrent high-grade glioma. Sci Transl Med 2016, 8, 341ra375, doi:10.1126/scitranslmed.aad9784.
- Cloughesy, T.F.; Petrecca, K.; Walbert, T.; Butowski, N.; Salacz, M.; Perry, J.; Damek, D.; Bota, D.; Bettegowda, C.; Zhu, J.J.; et al. Effect of Vocimagene Amiretrorepvec in Combination With Flucytosine vs Standard of Care on Survival Following Tumor Resection in Patients With Recurrent High-Grade Glioma: A Randomized Clinical Trial. JAMA Oncol 2020, 6, 1939-1946, doi:10.1001/jamaoncol.2020.3161.
- Hogan, D.J.; Zhu, J.J.; Diago, O.R.; Gammon, D.; Haghighi, A.; Lu, G.; Das, A.; Gruber, H.E.; Jolly, D.J.; Ostertag, D. Molecular Analyses Support the Safety and Activity of Retroviral Replicating Vector Toca 511 in Patients. Clin Cancer Res 2018, 24, 4680-4693, doi:10.1158/1078-0432.CCR-18-0619.
"Please see the attachment."

Round 2
Reviewer 2 Report
Comments and Suggestions for Authors
The Authors responses to the concerns raised are satisfactory. Hence, I recommend the manuscript for publication.